# Demographic characteristics, clinical symptoms, biochemical markers and probability of occurrence of severe dengue: A multicenter hospital-based study in Bangladesh

Jingli Yang[1,2], Abdullah Al Mosabbir[3], Enayetur Raheem[3], Wenbiao Hu[1]*, Mohammad Sorowar Hossain[3,4]*

1 Ecosystem Change and Population Health Research Group, School of Public Health and Social Work, Queensland University of Technology, Brisbane, Australia, 2 College of Earth and Environmental Sciences, Lanzhou University, Lanzhou, China, 3 Department of Emerging and Neglected Diseases, Biomedical Research Foundation, Dhaka, Bangladesh, 4 School of Environment and Life Sciences, Independent University, Dhaka, Bangladesh

* w2.hu@qut.edu.au (WH); sorowar.hossain@brfbd.org (MSH)

**Data Availability Statement:** All data generated or analyzed during this study are included in this

## Abstract

Establishing reliable early warning models for severe dengue cases is a high priority to facilitate triage in dengue-endemic areas and optimal use of limited resources. However, few studies have identified the complex interactive relationship between potential risk factors and severe dengue. This research aimed to assess the potential risk factors and detect their high-order combinative effects on severe dengue. A structured questionnaire was used to collect detailed dengue outbreak data from eight representative hospitals in Dhaka, Bangladesh, in 2019. Logistic regression and machine learning models were used to examine the complex effects of demographic characteristics, clinical symptoms, and biochemical markers on severe dengue. A total of 1,090 dengue cases (158 severe and 932 non-severe) were included in this study. Dyspnoea (Odds Ratio [OR] = 2.87, 95% Confidence Interval [CI]: 1.72 to 4.77), plasma leakage (OR = 3.61, 95% CI: 2.12 to 6.15), and hemorrhage (OR = 2.33, 95% CI: 1.46 to 3.73) were positively and significantly associated with the occurrence of severe dengue. Classification and regression tree models showed that the probability of occurrence of severe dengue cases ranged from 7% (age >12.5 years without plasma leakage) to 92.9% (age ≤12.5 years with dyspnoea and plasma leakage). The random forest model indicated that age was the most important factor in predicting severe dengue, followed by education, plasma leakage, platelet, and dyspnoea. The research provides new evidence to identify key risk factors contributing to severe dengue cases, which could be beneficial to clinical doctors to identify and predict the severity of dengue early.

published article and its Supporting information files.

**Funding:** Field-level data collection was partially funded by Techno Drug Ltd, Bangladesh to MSH, grant number Techno-2019-01. The funders had no role in study design, data collection and analysis, decision to publish, or preparation of the manuscript.

**Competing interests:** The authors report there are no competing interests to declare.

## Author summary

Dengue is a mosquito-borne viral infection mostly in warm and tropical regions, which has been listed as one of the top ten global health threats by the WHO. Among neglected tropical diseases, the mortality of dengue is on the rise. Severe dengue (typically manifested by bleeding, organ dysfunction, and plasma leakage) has become a leading cause of hospitalization for children and adults. There is a higher risk of death if severe dengue cases are not appropriately managed. Therefore, finding biomarkers that can reliably predict the development of severe dengue in symptomatic individuals is one of the main focuses of current research efforts. We found that dyspnoea, plasma leakage, and hemorrhage were the independent risk factors of severe dengue. The predictive probability of occurrence of severe dengue achieved 92.9% among people aged $\leq$ 12.5 years with dyspnoea and plasma leakage. Establishing an early warning system for severe dengue based on these factors is essential for triaging in endemic areas. The findings of this study identified possible combinations of severe dengue, which would provide enhanced insight into clinical management and inform prevention programming for severe dengue.

## Introduction

Dengue, a global public health concern, is a mosquito-borne viral infection mostly in warm and tropical regions [1]. Dengue is transmitted through a human-mosquito-human cycle, with *Aedes aegypti* as the primary vector and followed by *Aedes albopictus* [2]. Dengue cases reported by the World Health Organization (WHO) have increased 10-fold during the past 20 years, from 0.5 million cases in 2000 to 2.4 million cases in 2010 and 5.2 million cases in 2019 [3]. Dengue is the only neglected tropical disease whose mortality rose from 1990 to 2019 [4], which is likely due to the interaction of climate change [5], traffic [6], population density [7], extreme poverty, and inadequate sanitation [8]. Dengue has been listed as one of the top ten global health threats by the WHO in 2019, which was confirmed by recent outbreaks in many countries [1], such as the 2019 dengue outbreak in Bangladesh [9]. Salje estimated that about a quarter of Bangladesh's population (about 40 million) was infected with dengue, with an average of 2.4 million infections per year [10]. In severe dengue, which may cause bleeding, organ dysfunction, and plasma leakage, there is a higher risk of death if it is not treated correctly, which has become a leading cause of hospitalization and death for children and adults [3]. As reported by Stanaway, dengue caused about 1.14 million disability-adjusted life-years in 2013 [11].

There is no specific antiviral therapy for dengue infection, but the symptoms can be managed [1]. According to the WHO, coordinated processes for early detection, classifying, treating, and referring severe dengue lower mortality to less than 1% [3]. Vector control is the main strategy for dengue control [12], but recent studies show that existing control measures have achieved little in curbing the rising incidence of dengue infection globally [13]. Finding biomarkers that can reliably predict the development of severe dengue in symptomatic individuals is one of the main focuses of current research efforts [6]. During the transition from the febrile period to the critical period, WHO highlights several clinical warning signs as potential signs of impending exacerbation [14]. However, predictive models using clinical warning signs are lacking [14]. Establishing an early warning system for severe dengue based on these factors is essential for triaging in endemic areas and optimizing resource utilization [6].

Our previous study used the data to initially explore clinical symptoms of severe dengue among children during the 2019 outbreak in Bangladesh and suggested dengue was more

severe in children, mainly gastrointestinal symptoms [9]. However, this study has not explored the high-order and combinative effect on different features. The classification and regression tree (CART) and random forest (RF) models are nonparametric statistical methods that do not rely on assumptions about data distribution to handle multiple independent variables interacting [15]. In this study, we aimed to use both CART and RF to explore the possible combination patterns of demographic characteristics, clinical symptoms, and biochemical markers on severe dengue, eventually providing opportunities for early treatment and prevention of severe dengue and providing a reference for clinicians.

## Materials and methods

### Ethics statement

All participants gave written informed consent. The protocol of the survey was approved by the Ethical Review Committee (ERC) of the Biomedical Research Foundation (Memo no: BRF/ERB/2019/017). In case of child participants, formal consent was obtained from the parent/guardian. This study is reported as per the Strengthening the Reporting of Observational Studies in Epidemiology (STROBE) guideline for cross-sectional studies (S1 STROBE Statement).

### Study site and data collection

This study was conducted in the Dhaka megacity (population = ~20M) of Bangladesh where a massive dengue outbreak occurred in 2019. We used convenience sampling technique to enroll hospitalized dengue cases from eight hospitals (five government and three private hospitals) between 15 August 2019 and 30 September 2019. The diagnosis was confirmed serologically by the positivity of antibodies against the dengue nonstructural glycoprotein 1 (NS1). Trained medical students and doctors collected the data via face-to-face interviews with a structured questionnaire. Data were collected at a one-time point at the convalescent phase to record all the clinical and laboratory complications patients developed during their hospital stay. The questionnaire was developed based on previously published literature and discussions with interdisciplinary teams (e.g., epidemiologists, clinicians, and public health workers), and details of the investigation can be found in the published literature [9]. A total of 1,283 patients with dengue were investigated by a structured questionnaire. Finally, 1,090 dengue cases were included in this study after excluding the incomplete data on age, sex, and type of dengue (missing rate: 14.18%) (S1 Fig).

Standardized questionnaires (S1 Questionnaire) were used to collect information on demographic characteristics, clinical symptoms, and biochemical markers. Demographic characteristics included age (categorized as <18, 18–39, or ≥ 40 years), sex (dichotomized as men or women), education (categorized as illiterate or children, primary, secondary and high secondary school, or graduate), monthly income (categorized as <1,5000, 1,5000–2,5000, 2,5000–50,000, or ≥ 50,000 Bangladeshi taka), type of residence (categorized as flat, single storied house and other, or tin shade house), history of infection (categorized as dengue, chikungunya, or none), and history of comorbidity (dichotomized as yes or no). Clinical symptoms were asked about the presence of these symptoms [16], which included fever, rash, joint pain, dehydration, itchiness, plasma leakage, hemorrhage (dichotomized as yes or no), and muscle pain, vomiting, headache, decreased appetite, abdominal pain, cough, back pain, dyspnoea, lethargy (categorized as yes, no, or can't remember). In addition, biochemical markers included platelet counts, hemoglobin, white blood cell (WBC, dichotomized as reduced or within normal), and alanine transaminase (ALT), aspartate transaminase (AST) (categorized as raised, within normal, or not done). As per the National Guideline for Clinical Management of Dengue,

symptomatic dengue cases were initially divided into three groups: dengue fever (group A), dengue fever with warning signs (group B), and severe dengue (group C). In this study, group A and group B were treated as the non-severe dengue group, while group C was treated as the severe dengue group [17].

## Statistical analyses

Descriptive statistics were employed to describe the counts (percentages) of demographic characteristics, clinical symptoms, and biochemical markers grouped by severe dengue and non-severe dengue. Chi-square tests were used for group comparisons. Pairwise spearman's rank correlations were used to detect the inter-correlations between demographic characteristics, clinical symptoms, and biochemical markers. In this study, the missing values were less than 5% among most of the covariates. We used a listwise deletion method to deal with the missing data [18]. The logistic regression model was used to explore the associations between demographic characteristics, clinical symptoms, biochemical markers, and the risk of severe dengue. We used the variance inflation factor (VIF) to estimate multicollinearity in the multiple logistic regression models.

The CART model was used to examine the combinative effects of demographic characteristics, clinical symptoms, and biochemical markers on severe dengue and to identify the high-order no-linear pattern of developing severe dengue. To get the probability of severe dengue, dengue was treated as a scale factor in CART model. CART model was used to split the data into two parts that were as homogeneous as possible for the dependent variable. We conducted split-sample validation by using random assignment, which allowed the model to be generated using a training sample (80%) and tested on a hold-out sample (20%) [19]. For the minimum number of cases, we set the parent node to 30 and the child node to 10. To avoid overfitting the model, we also pruned the tree. Other parameters were set as default.

The RF model was used to find more important factors among demographic characteristics, clinical symptoms, and biochemical markers of severe dengue. In this study, the method of mean decreased accuracy was applied to compute the feature importance on permuted out-of-bag (OOB) samples [20]. Higher mean decreased accuracy implied more important features. Data were analyzed using R version 4.1.3 software (R Foundation for Statistical Computing, Vienna, Austria). The CART model was employed in SPSS 27.0. (IBM, Armonk, NY, USA). A two-sided $P$ value $< 0.05$ were considered statistically significant, and the RF model was performed using the 'randomForest' and 'rfPermute' packages in R software [20].

## Results

### Basic characteristics of the study participants with dengue

A total of 1,090 dengue cases (158 severe and 932 non-severe) were included in this study. Overall, 652 (59.8%) were men, 553 (50.7%) aged 18–39 years. Compared to patients with non-severe dengue, patients with severe dengue were younger (aged <18 years) and more likely to be illiterate. There was no statistically significant difference between severe and non-severe dengue in other demographic characteristics (Table 1).

Out of 1,090 dengue patients, 1,034 (94.9%) had fever, 691 (63.4%) muscle pain, 837 (76.8%) vomiting, 901 (82.7%) headache, 631 (57.9%) abdominal pain, 651 (59.7%) back pain, and 869 (79.7%) decreased appetite. Compared to patients with non-severe dengue, patients with severe dengue were more likely to have abdominal pain ($P$ value = 0.04) (S1 Table).

Among 1,090 dengue patients, 972 (89.2%) appeared low platelet level, 315 (28.9%) had low hemoglobin level, 404 (37.1%) had low WBC, 183 (16.8%) raised ALT, and 148 (13.6%) raised AST (S2 Table).

**Table 1. Socio-demographic characteristics of the dengue cases\*.**

| | Overall | Non-severe dengue | Severe dengue | *P* value |
|---|---|---|---|---|
| | (N = 1090) | (N = 932) | (N = 158) | |
| **Sex** | | | | 0.08 |
| Men | 652 (59.8) | 568 (60.9) | 84 (53.2) | |
| Women | 438 (40.2) | 364 (39.1) | 74 (46.8) | |
| **Age group (years)** | | | | <0.01 |
| <18 | 318 (29.2) | 226 (24.2) | 92 (58.2) | |
| 18–39 | 553 (50.7) | 507 (54.4) | 46 (29.1) | |
| 40 and above | 219 (20.1) | 199 (21.4) | 20 (12.7) | |
| **Education** | | | | <0.01 |
| Illiterate or Children | 284 (26.1) | 223 (23.9) | 61 (38.6) | |
| Primary | 339 (31.1) | 303 (32.5) | 36 (22.8) | |
| Secondary & Higher Secondary School | 306 (28.1) | 264 (28.3) | 42 (26.6) | |
| Graduate | 112 (10.3) | 99 (10.6) | 13 (8.2) | |
| Missing | 49 (4.5) | 43 (4.6) | 6 (3.8) | |
| **Monthly income (BDT)\*** | | | | 0.36 |
| < 15000 | 342 (31.4) | 290 (31.1) | 52 (32.9) | |
| 15000–25000 | 404 (37.1) | 353 (37.9) | 51 (32.3) | |
| 25000–50000 | 206 (18.9) | 169 (18.1) | 37 (23.4) | |
| 50000 and above | 71 (6.5) | 60 (6.4) | 11 (7.0) | |
| Missing | 67 (6.1) | 60 (6.4) | 7 (4.4) | |
| **Type of residence** | | | | 0.99 |
| Flat | 540 (49.5) | 461 (49.5) | 79 (50.0) | |
| Single storied house | 125 (11.5) | 107 (11.5) | 18 (11.4) | |
| Tin shade house and slum | 384 (35.2) | 327 (35.1) | 57 (36.1) | |
| Missing | 41 (3.8) | 37 (4.0) | 4 (2.5) | |
| **History of infection** | | | | 0.66 |
| Dengue | 24 (2.2) | 21 (2.3) | 3 (1.9) | |
| Chikungunya | 116 (10.6) | 102 (10.9) | 14 (8.9) | |
| None | 903 (82.8) | 767 (82.3) | 136 (86.1) | |
| Missing | 47 (4.3) | 42 (4.5) | 5 (3.2) | |
| **History of comorbidity** | | | | 0.74 |
| Yes | 173 (15.9) | 146 (15.7) | 27 (17.1) | |
| No | 917 (84.1) | 786 (84.3) | 131 (82.9) | |

BDT, Bangladeshi taka.

\* Values are presented as n (%).

## Association of demographic characteristics, clinical symptoms, and biochemical markers with the risk of severe dengue

**Logistic regression model.** In the crude logistic regression model, age, education, headache, abdominal pain, back pain, dyspnoea, plasma leakage, hemorrhage, and WBC were statistically significantly associated with severe dengue (S3 Table). Pairwise spearman's rank correlation showed that the coefficients among significant factors in the crude logistic regression ranged from -0.16 to 0.32 (S2 Fig). The factors significantly associated with the crude logistic regression model were included in the multiple logistic regression model. All VIF values were under 10 in the multiple logistic regression model, implying no substantial issue in multicollinearity (S4 Table). The results showed that patients with hemorrhage (OR = 2.33, 95% CI:

**Table 2. Association of demographic characteristics, clinical symptoms, and biochemical markers with the risk of severe dengue in multiple logistic regression model.**

| | No of non-severe dengue | No. of severe dengue | OR | 95% CI for OR | | P value |
|---|---|---|---|---|---|---|
| | | | | Lower | Upper | |
| **Age group (years)** | | | | | | |
| <18 | 226 | 92 | 1.00 (Ref.) | 1.00 | 1.00 | 1.00 |
| 18–39 | 507 | 46 | 0.20 | 0.11 | 0.36 | <0.01 |
| 40 and above | 199 | 20 | 0.17 | 0.08 | 0.35 | <0.01 |
| **Education** | | | | | | |
| Illiterate or Children | 223 | 61 | 1.00 (Ref.) | 1.00 | 1.00 | 1.00 |
| Primary | 303 | 36 | 0.47 | 0.27 | 0.85 | 0.01 |
| Secondary & Higher Secondary School | 264 | 42 | 0.88 | 0.46 | 1.66 | 0.69 |
| Graduate | 99 | 13 | 0.80 | 0.33 | 1.96 | 0.63 |
| **Headache** | | | | | | |
| No | 135 | 36 | 1.00 (Ref.) | 1.00 | 1.00 | 1.00 |
| Yes | 782 | 119 | 0.97 | 0.54 | 1.73 | 0.92 |
| **Abdominal pain** | | | | | | |
| No | 397 | 53 | 1.00 (Ref.) | 1.00 | 1.00 | 1.00 |
| Yes | 527 | 104 | 0.95 | 0.59 | 1.54 | 0.84 |
| **Back pain** | | | | | | |
| No | 320 | 69 | 1.00 (Ref.) | 1.00 | 1.00 | 1.00 |
| Yes | 571 | 80 | 0.64 | 0.39 | 1.06 | 0.09 |
| **Dyspnoea** | | | | | | |
| No | 752 | 95 | 1.00 (Ref.) | 1.00 | 1.00 | 1.00 |
| Yes | 163 | 60 | 2.87 | 1.72 | 4.77 | <0.01 |
| **Plasma leakage** | | | | | | |
| No | 789 | 98 | 1.00 (Ref.) | 1.00 | 1.00 | 1.00 |
| Yes | 106 | 53 | 3.61 | 2.12 | 6.15 | <0.01 |
| **Hemorrhage** | | | | | | |
| No | 682 | 84 | 1.00 (Ref.) | 1.00 | 1.00 | 1.00 |
| Yes | 216 | 56 | 2.33 | 1.46 | 3.73 | <0.01 |
| **White blood cell** | | | | | | |
| Within normal | 518 | 106 | 1.00 (Ref.) | 1.00 | 1.00 | 1.00 |
| Reduced | 363 | 41 | 0.52 | 0.32 | 0.85 | 0.01 |

1.46 to 3.73), plasma leakage (OR = 3.61, 95% CI: 2.12 to 6.15), and dyspnoea (OR = 2.87, 95% CI: 1.72 to 4.77) were positively associated with the risk of severe dengue (Table 2). The hemorrhage mainly occurred in the gum, hematochezia, and menorrhagia, and plasma leakage mainly in abdomen and chest (S5 Table). Furthermore, severe dengue in the non-DSS group was significantly higher in participants with plasma leakage or hemorrhage than in those without plasma leakage or hemorrhage (S6 Table). Compared to illiterate, primary education was associated with a lower risk of severe dengue (OR = 0.47, 95% CI: 0.27 to 0.85). Compared to patients aged <18 years, patients aged 18–39 years (OR = 0.20, 95% CI: 0.11 to 0.36) and >40 years (OR = 0.17, 95% CI: 0.08 to 0.35) had less likely to have severe dengue (Table 2).

## CART models

**Model I: Demographic characteristics.** The CART analysis showed that age (threshold: 11.5 years old) was the main associated factor, explaining 43.1% of severe dengue. S3 Fig shows that the prevalence of severe dengue was 55.2% for women aged between 1.9 and 11.5

years and with monthly income <25,000 BDT. Among patients aged >51.5 years, a higher education level (over primary education) and comorbidity explained 42.9% of severe dengue.

**Model II: Clinical symptoms.**   The prevalence of severe dengue was 76.5% for patients with plasma leakage, dyspnoea, and hemorrhage (S4 Fig). Patients without plasma leakage had 10.7% of severe dengue.

**Model III: Biochemical markers.**   S5 Fig shows that the prevalence of severe dengue was 54.3% for patients whose lowest platelet levels were between $71.5^{*}10^{9}$/L and $78.5^{*}10^{9}$/L. The lowest platelet (threshold: $71.5^{*}10^{9}$/L) was associated with factors for non-severe dengue (93.1%).

**Model IV: Combination of demographic characteristics, clinical symptoms, and biochemical markers.**   The final CART model showed that age, dyspnoea, plasma leakage, and lowest platelet contributed to the predictive power of the CART algorithm. The prevalence of severe dengue across nodes ranged from 7% (age >12.5 years without plasma leakage) to 92.9% (age ≤ 12.5 years with dyspnoea and plasma leakage). The prevalence of severe dengue was 60.0% among people aged > 12.5 years with plasma leakage, and the lowest platelet level was more than $126.5^{*}10^{9}$/L (Fig 1).

## RF model

Fig 2 shows that age was the most important variable (mean decrease accuracy = 23.6%) in predicting severe dengue, followed by education (11.6%), plasma leakage (9.9%), platelet (8.5%), and dyspnoea (7.2%) (S7 Table). The accuracy of RF model was 86.5%, and the OOB estimate of error rate was 13.5%.

## Discussion

In this study, we have assessed the potential risk factors of severe dengue during the largest outbreak in Bangladesh in 2019 and detected the high-order combinative effect of demographic characteristics, clinical symptoms, and biochemical markers for developing severe dengue. We found that dyspnoea, plasma leakage, and hemorrhage were the independent risk factors of severe dengue. The predictive probability of occurrence of severe dengue achieved 92.9% among people aged ≤ 12.5 years with dyspnoea and plasma leakage. Furthermore, age was the most crucial variable in predicting severe dengue, followed by education, plasma leakage, platelet, and dyspnoea.

Our research found that age was associated with the risk of severe dengue. A piece of evidence from the Global Burden of Disease Study 2019 showed that the age-standardized mortality rate and disability-adjusted life years for dengue in children were higher than in older people [4], which might relate to fragile capillaries in children than adults [21]. As previous studies reported, we found that the prevalence of severe dengue increased in specific populations, such as the older with comorbidities [22]. However, the degree of severity of dengue may differ from the different diseases [23], the type of comorbidity should be investigated in the future study. Additionally, our study found that patients who received higher educational levels were less likely to be infected with severe dengue, possibly because patients with higher educational attainment may be better aware of dengue infection and seek professional help promptly [24]. Moreover, women were more likely to be infected with severe dengue than men in this study. Results from several studies supported that men were likely to be infected with dengue in dengue epidemics, but women were more likely associated with severe dengue [25]. Sex-specific may be related to differences in visit time and type of care [26]. Therefore, educational public health campaigns might target lower-income areas, individuals with comorbidities, and families with children.

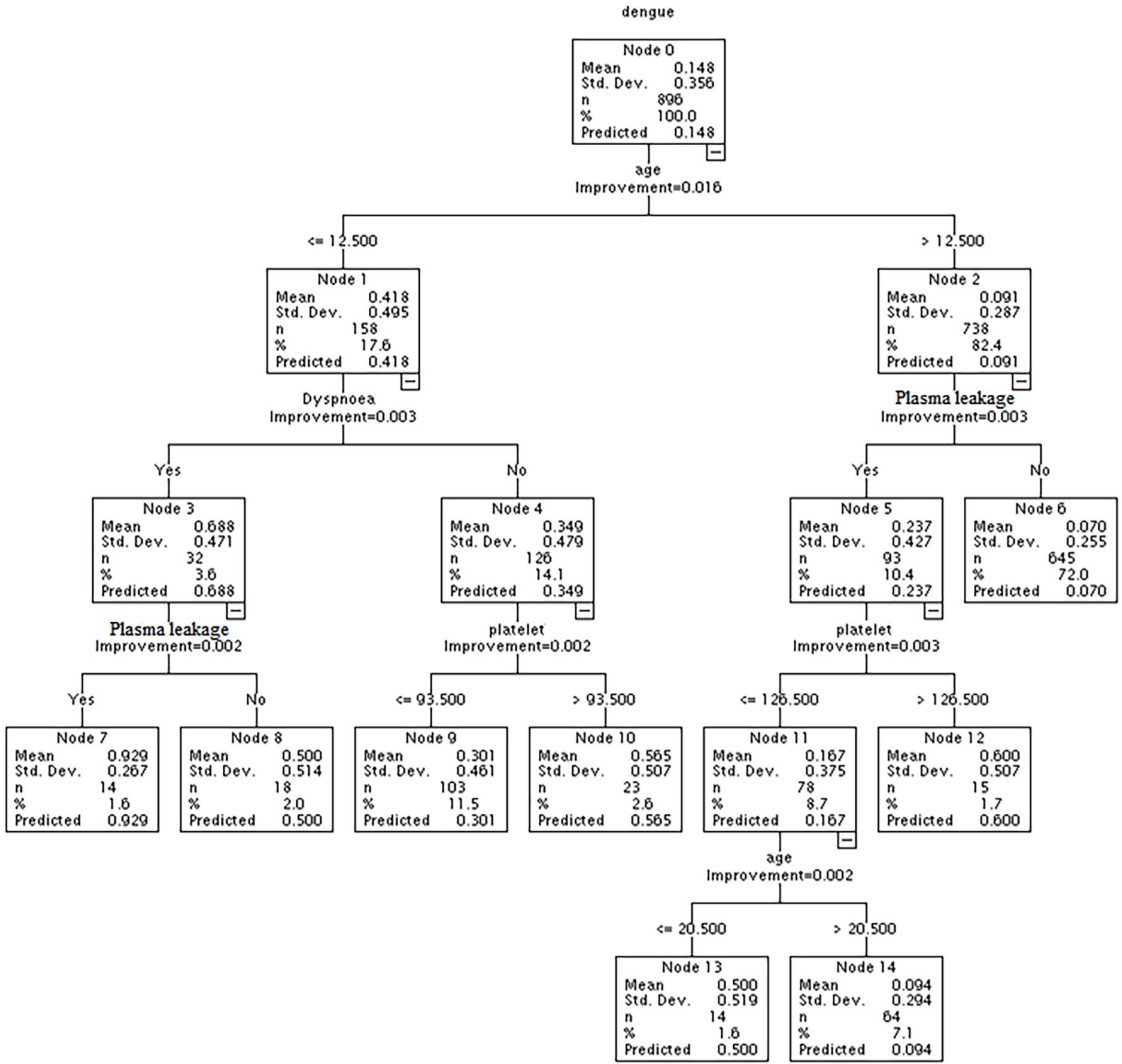

**Fig 1. The probability of demographic characteristics, clinical symptoms, and biochemical markers for developing severe dengue in the classification and regression tree (CART) model.**

The most common characteristic of severe dengue is plasma leakage followed by hemorrhage [27], which supported our findings. Interestingly, we found that plasma leakage occurred mainly in children, which may be related to the lower leakage threshold in children than in adults [28]. Furthermore, we found that plasma leakage occurred mainly in the abdomen and chest, and that people with ascites and pleural effusion were more likely to develop severe dengue. Plasma leakage is an important outcome for dengue, as most complications occurred in this group. Identifying all plasma leakage is an important factor in preventing complications, but neither WHO classification fully covers the plasma leakage subgroup [29,30]. In addition, the definition and development of criteria for diagnosing plasma leakage have long been neglected, resulting in challenging and underreporting of plasma leakage. It is recommended that standardization of diagnosis and reporting of plasma leakage should be a research priority for dengue [31]. Moreover, we found that patients with hemorrhage were more likely to develop severe dengue than those without hemorrhage. Hemorrhage during dengue infection ranges from self-limiting epistaxis and bleeding gums to life-threatening

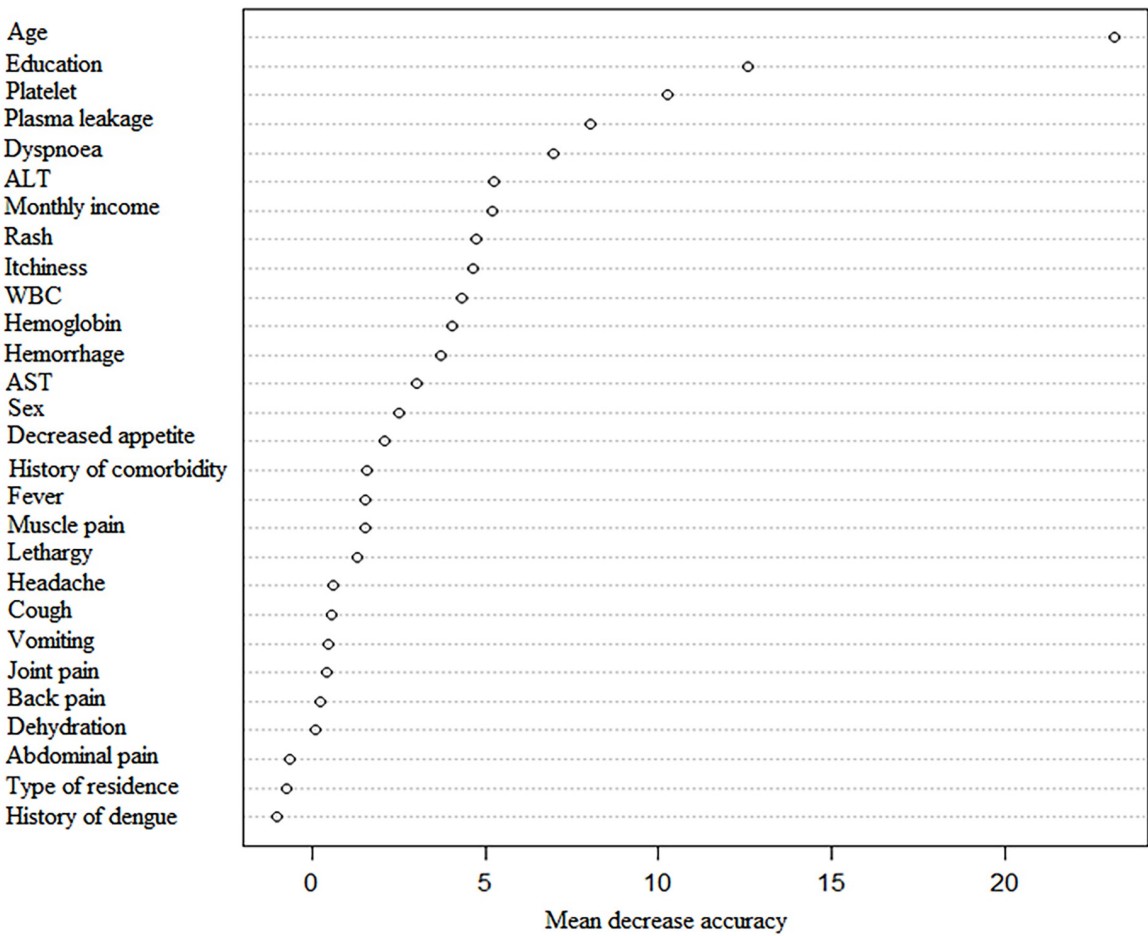

**Fig 2. The random forest (RF) model identified the important predictors for predicting severe dengue.**

gastrointestinal bleeding. In addition, previous studies have found that hemorrhage is often attributed to thrombocytopenia [32], which is consistent with our findings. Thrombocytopenia and hemorrhage in dengue may result from cytokine-induced immune dysfunction and platelet destruction following the binding of dengue-specific antibodies to virus-infected platelets [32]. Lower platelet counts were more common in severe dengue [33] and were considered a risk factor for bleeding [34]. During dengue fever, platelet counts declined until fever subsides, then recovered rapidly [35]. Furthermore, we found that dyspnoea was a vital sign of severe dengue. Published research showed a significant association of dyspnoea with shock, which may result from fluid overload due to pulmonary congestion caused by capillary leakage [36]. Early recognition and treatment of dyspnea are essential for preventing severe dengue.

Furthermore, clinicians should be vigilant in identifying dengue patients, as prompt identification and early treatment of dengue remain the cornerstone of reducing morbidity and mortality [37]. In recent years, climate change, environmental pollution, poor health systems, and population growth have been exacerbating the dengue outbreaks in Bangladesh [38]. Clinicians should be aware of the different presentations of severe dengue in children versus adults and carefully evaluate the children for plasma leakage, which may be more challenging to identify than severe bleeding. As reported, if detected early and with appropriate medical care, the mortality rate of dengue was less than 1% in Bangladesh [39]. To reduce mortality from severe dengue, clinicians need to be trained in managing severe dengue [40] and focus

on the high-risk populations of dengue-affected areas [41]. The results of our study identified possible combinations of severe dengue, which would provide an essential reference for the early identification, and treatment of severe dengue fever, especially for patients without DSS.

In the research, we used machine learning algorithms, including CART and RF, to find key risk factors and to identify the combined effect of risk factors on all severe dengue cases, which include cases with key symptoms such as hemorrhage and plasma leakage etc, and without these key symptoms. Our results will provide very useful warning signals for all dengue cases for clinical doctors. However, there are some limitations to this study. First, our results need to be confirmed by further prospective studies due to the cross-sectional design. Some risk factors, especially biomarkers and clinical symptoms, may be affected by the process of dengue. Second, the biomarkers, such as platelet counts, were only measured once in this study. However, the levels of biomarkers were more likely to be affected by the status and treatment of diseases. The changes in biomarkers (particularly platelet counts) may carry meaningful information, and developing models incorporating repeatedly measured longitudinal data will improve the accuracy of risk prediction. However, results from the present study have provided new evidence on the high-order interactive effects of demographic characteristics, clinical symptoms, biochemical markers, and severe dengue. The finding will provide enhanced insight into clinical treatment and inform prevention programming for severe dengue. Third, due to regional differences, the results of this study may not be generalizable to other regions. Fourth, patients were interviewed during the convalescent phase; therefore, some recall biases are inevitable in this study. However, this should be negligible as most of the dengue patients reach convalescent within 7–10 days of starting symptoms [29]. Lastly, given that the risk of developing severe dengue could be associated with secondary infections [42]. The study did not explore this relationship because of lack of detailed data on IgG (an indicator of secondary infection) and dengue virus serotypes.

In conclusion, combined with socio-environmental factors, early recognition and treatment of dyspnoea, plasma leakage, and hemorrhage are vital to the prevention of severe dengue. The research provides new evidence to identify key risk factors that contribute to severe dengue cases, which would add new evidence for the early identification and treatment of severe dengue fever. Future research is required to further explore the progression and shifting from non-severe dengue to severe dengue based on the research finding.

## Supporting information

**S1 STROBE Statement. Checklist of items that should be included in reports of cross-sectional studies.**
(DOCX)

**S1 Questionnaire. Data collection form.**
(DOCX)

**S1 Fig. Flowchart of participant selection.** Abbreviation: DMCH: Dhaka Medical College Hospital; KuGH: Kurmitola General Hospital; MuMCH: Mugda Medical College Hospital; PMCH: Popular Medical College Hospital; DrSIMCH: Dr Sirajul Islam Medical College Hospital; ShMCH: Suhrawardy Medical College Hospital; MHSHMC: MH Samorita Hospital and Medical College; SSMC: Sir Salimullah Medical College Hospital.
(TIF)

**S2 Fig. The correlations between demographic characteristics, clinical symptoms, and biochemical markers were selected in the single logistic regression model.**
(TIF)

**S3 Fig. The patterns of demographic characteristics for developing severe dengue in the classification and regression tree (CART) model.**
(TIF)

**S4 Fig. The patterns of clinical symptoms for developing severe dengue in classification and regression tree (CART) model.**
(TIF)

**S5 Fig. The patterns of biochemical markers for developing severe dengue in classification and regression tree (CART) model.**
(TIF)

**S1 Table. Clinical features of the patients grouped by severity of dengue**\*. \* Values are presented as n (%).
(DOCX)

**S2 Table. Laboratory findings of the patients grouped by severity of dengue**\*. Abbreviation: WBC, white blood cell; ALT, alanine transaminase; AST, aspartate transaminase. \* Values are presented as n (%).
(DOCX)

**S3 Table. Association of demographic characteristics, clinical symptoms, and biochemical markers with the risk of severe dengue in crude logistic regression model.** Abbreviation: WBC, white blood cell; ALT, alanine transaminase; AST, aspartate transaminase.
(DOCX)

**S4 Table. Collinearity analysis (variance inflation factor, VIFs) in multiple logistic regression model.**
(DOCX)

**S5 Table. Clinical features of the patients grouped by severity of dengue and the site of haemorrhage and plasma leakage**\*. \* Values are presented as n (%).
(DOCX)

**S6 Table. Clinical features of the patients grouped by severity of dengue and DSS**\*. Abbreviation: DSS, dengue shock syndrome. \* Values are presented as n (%).
(DOCX)

**S7 Table. The random forest model identified the important predictors on the prediction of severe dengue.** Abbreviation: WBC, white blood cell; ALT, alanine transaminase; AST, aspartate transaminase.
(DOCX)

## Acknowledgments

We would like to acknowledge the following persons for supporting data collection: Mahbubul H Siddiqee (Biomedical Research Foundation, Bangladesh); Professor Robed Amin and Associate Professor Dr. Syed Ghulam Mogni Mowla(Dhaka Medical College Hospital), Lt Col ABM Belayet Hossain and Farah Noor (Kurmitola General Hospital), Associate Professor Sudip Ranjan Deb (Mugda Medical College Hospital), Professor HAM Nazmul Ahsan and Professor Quazi Tarikul Islam (Popular Medical College Hospital) Associate Professor Sabrina Yesmin (Dr. Sirajul Islam Medical College Hospital), Associate Professor Nazmul Ahsan and Associate Professor Mohammad Rafiqul Islam (Suhrawardy Medical College Hospital), Professor Syeda Afroza (Samaritan Hospital and Medical College), and Associate Professor

Amiruzzaman Sir Salimullah Medical College Hospital. Besides, Jingli Yang would thank the support from the Queensland University of Technology and the China Scholarship Council (CSC).

## Author Contributions

**Conceptualization:** Jingli Yang, Wenbiao Hu, Mohammad Sorowar Hossain.

**Data curation:** Jingli Yang, Abdullah Al Mosabbir, Enayetur Raheem, Wenbiao Hu, Mohammad Sorowar Hossain.

**Formal analysis:** Jingli Yang, Wenbiao Hu.

**Funding acquisition:** Mohammad Sorowar Hossain.

**Investigation:** Jingli Yang, Abdullah Al Mosabbir, Enayetur Raheem, Wenbiao Hu, Mohammad Sorowar Hossain.

**Methodology:** Jingli Yang, Wenbiao Hu.

**Project administration:** Wenbiao Hu, Mohammad Sorowar Hossain.

**Resources:** Wenbiao Hu, Mohammad Sorowar Hossain.

**Supervision:** Wenbiao Hu, Mohammad Sorowar Hossain.

**Validation:** Jingli Yang, Wenbiao Hu.

**Visualization:** Jingli Yang.

**Writing – original draft:** Jingli Yang.

**Writing – review & editing:** Jingli Yang, Abdullah Al Mosabbir, Enayetur Raheem, Wenbiao Hu, Mohammad Sorowar Hossain.

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
