## [Decision Letter · Decision Letter 0]

27 Oct 2022

Dear Dr. Hossain,

Thank you very much for submitting your manuscript "Demographic characteristics, clinical symptoms, biochemical markers and probability of occurrence of severe dengue: A multicenter hospital-based study in Bangladesh" for consideration at PLOS Neglected Tropical Diseases. As with all papers reviewed by the journal, your manuscript was reviewed by members of the editorial board and by several independent reviewers. In light of the reviews (below this email), we would like to invite the resubmission of a significantly-revised version that takes into account the reviewers' comments. 

We cannot make any decision about publication until we have seen the revised manuscript and your response to the reviewers' comments. Your revised manuscript is also likely to be sent to reviewers for further evaluation.

Sincerely,

Marc Choisy

Academic Editor

Abdallah Samy

Section Editor

Editor comments: The 3 reviewers found the study has potential but all of them raised major issues regarding both the analyses and the writing. We are happy to consider a resubmission of your manuscript if you think you will be able to address all the reviewers' comments.

Reviewer's Responses to Questions

**Key Review Criteria Required for Acceptance?**

**Methods**

-Are the objectives of the study clearly articulated with a clear testable hypothesis stated?

-Is the study design appropriate to address the stated objectives?

-Is the population clearly described and appropriate for the hypothesis being tested?

-Is the sample size sufficient to ensure adequate power to address the hypothesis being tested?

-Were correct statistical analysis used to support conclusions?

-Are there concerns about ethical or regulatory requirements being met?

Reviewer #1: This study is designed well but there is a need to improve further on the following parts:

1. Sample size calculation

2. Definition of severe and non-severe dengue

3. How the collinearity assessment was done between different variables

4. Clear explanation on co-morbidity, hemorrhage and plasma leakage

Reviewer #2: This is an exploratory analysis, so there is no specific hypothesis.

This is a large population from Bangladesh including both severe and non-severe dengue cases, which is appropriate for exploring variables that may differentiate between severe and non-severe dengue. 

Can the authors please state how dengue was diagnosed? Did they use PCR, serology, both? Do they have data on which serotypes were causing infection?

Since this is a cross-sectional study, I expect only 1 time point was measured. However, the author's should state this explicitly. If only one time point was measured, I think the authors should report and control for days post symptom onset in their models. Per the data form shared in the supplementary material, it looks like symptom onset day may have been collected. 

Additionally, can the authors explain how they defined dengue and non-severe dengue? In the methods, they explain that they dichotomized the data as non-severe and severe dengue, and then they cite the WHO criteria. This is confusing to me because, according to their models, the three variables with the strongest association with severe dengue (plasma leakage, hemorrhage, and dyspnea) are all part of the WHO criteria for severe dengue. If they used the WHO criteria to define severe dengue wouldn't it follow that the models would reflect the WHO criteria when identifying variables associated with severe dengue? Maybe I am misunderstanding how these models work, but then I think it would be helpful to walk the reader through this a bit more.

This is just a suggestion, but it might actually be informative to do an unsupervised clustering analysis where they just see which variables cluster together without considering whether the dengue is severe or not. This would remove the WHO definitions from the model and might allow the authors to just observe which variables cluster together and potentially independently confirm the WHO definitions.

Reviewer #3: This is a large and unique clinical cohort which includes 1,090 participants (932 non-severe patients and 158 severe patients). This study also includes both adult and pediatric participants which allows them to evaluate age-dependent differences in the development of severe dengue. 

The authors use multiple non-parametric measures to identify patterns in their data which have enabled them to quantify the clinical, demographic and laboratory parameters which increase the probability of developing severe dengue. 

Patient questionnaires were performed during the convalescent phase of dengue infection and therefore may not be sensitive or specific for the prognosis of severe dengue.

This is a single cohort study and it is unclear if these findings could be extrapolated to other dengue endemic areas which may have different epidemiological aspects or common co-infections such as malaria.

**Results**

-Does the analysis presented match the analysis plan?

-Are the results clearly and completely presented?

-Are the figures (Tables, Images) of sufficient quality for clarity?

Reviewer #1: Results are presented well but there is need to improve further on:

1. Tabular presentation

2. Sentence struture

Reviewer #2: The analysis does match the analysis plan and the tables are clear, well structured, and helpful. 

The figures are a bit blurry, and the explanation of figure 1 seems to have a typo (line 220).

Reviewer #3: The presentation of the results from Figure 1 describing the CART model are difficult to interpret. Generating a figure which more clearly visualizes the mean and SD for each node would improve understanding of the results.

**Conclusions**

-Are the conclusions supported by the data presented?

-Are the limitations of analysis clearly described?

-Do the authors discuss how these data can be helpful to advance our understanding of the topic under study?

-Is public health relevance addressed?

Reviewer #1: There is no conclusion in the manuscript. It is suggested to include a paragraph to summarize the finding at the end of the discussion

Reviewer #2: Can the authors do a bit better job explaining what their study adds to the literature? For example, in line 238 - 239, they state that "The predictive probability of occurrence of severe dengue achieved 92.9% among people aged ≤ 12.5 years with dyspnoea and plasma leakage." But, if a clinician just follows the WHO criteria, then 100% of patients with plasma leakage would meet the criteria for severe dengue. The same is true for line 257, where they report that those with hemorrhage were 2.33 times more likely to develop severe dengue. But, if a clinician just follows WHO criteria, then 100% of patients with dengue and hemorrhage have severe dengue. So, I just don't really understand what the added value is of the model. Could the authors help readers understand that a bit better?

I think the associations between severe dengue and age and education levels are interesting and notable, but of course may not be generalizable to all cohorts. This limited generalizability should be noted. 

The authors make a few conclusions stating that their data provides "new evidence" (line 292) or "enhanced insight into clinical treatment" (line 294), but I am not sure that these conclusions are well justified. Can the authors explain what the new evidence is and how this will impact clinical treatment and prevention? It seemed that the clinical markers they report here are all consistent with the WHO criteria, so they seem to be confirming the WHO criteria rather then adding to them. The age and educational associations with severe dengue could potentially be helpful though. For example, educational public health campaigns might target lower income areas and families with children. 

In line 283, the authors state that "This study is the first to study severe dengue from a multi-index and multimethod perspective." Again, I am not sure this is quite true. There are a number of meta-analyses and cohort studies that have examined severe dengue and the authors may consider citing some of these. For example, Yaun et al. PLOS NTD, 2022. Sangkaew et al. Lancet ID, 2021. Paz-Bailey et al. JID 2022.

Reviewer #3: The authors state the clear public health relevance of evaluating risk factors associated with severe dengue given dengue remains a WHO top ten global health threat with increasing mortality rates. 

The authors do not fully describe the limitations of using study questionnaires during the convalescent phase of infection or discuss potential recall biases during interviews. Given that there was no association between primary and secondary dengue infections and severity, a well known risk factor for developing severe dengue, it would be important to discuss these challenges.

**Editorial and Data Presentation Modifications?**

Reviewer #1: (No Response)

Reviewer #2: Line 293: seems to have a typo of "n" 

Line 253: what do they mean by nursing? Is there some data to suggest that breastfeeding women are at higher risk for severe dengue? Do they know if many of the women in their cohort were breastfeeding? Since they have the data, it might be helpful to see if the sex differences were true in children and adults. If they only saw sex differences in adults, then potentially there is some hormonal component?

Reviewer #3: The quality of the tables and figures should be improved. Figure 1 and 2 are blurry and at times the wording is not clear.

**Summary and General Comments**

Reviewer #1: (No Response)

Reviewer #2: This is an exploratory analysis of a large cohort of hospitalized patients in Bangladesh. The study is strengthened by its large sample size of both severe and non-severe dengue cases. The authors report some interesting findings including the association of age and low education with severe dengue, and higher frequencies of plasma leakage in children. However, it would be helpful to know how dengue was diagnosed and how severe vs. non-severe dengue was determined. Additionally, I was confused by the models since it seemed that the variables that were strongly associated with severe dengue are part of the WHO criteria for severe dengue. Thus, I didn't understand the added value of the models. Can the authors expand on how these models add to the WHO criteria?

Reviewer #3: In this manuscript, Yang et al. report the results of a cross-sectional study in which they performed structured face-to-face interviews with a total 1,090 participants with dengue infection in Bangladesh. They looked specifically at patient demographics, laboratory findings, and clinical symptoms which were associated with participants who developed severe dengue based on the 2009 WHO criteria. 

The authors concluded that several factors were positively associated with severe dengue including dyspnea, plasma leakage and hemorrhage. They also applied a nonparametric statistic model in order to identify patterns in the data which increase the probability of developing severe dengue. This analysis revealed that age ( <= 12.5 yrs) was the most important factor in predicting severe dengue along with education, plasma leakage, platelet count and dyspnea. Taken together the authors state that this data may assist clinicians in the early identification and prediction of severe dengue cases. 

Strengths:

The size of the cohort and the number of severe dengue participants adds to our overall understanding of factors associated with severe dengue. The enrollment of young children also allows for more indepth analyses. The use of multiple non-parametric statistical analyses with similar findings strengths the results.

Weakness: 

The findings of this study generally coincide with the known associations of severe dengue which are part of the WHO classifications (thrombocytopenia, hemorrhage, plasma leakage). Children have also previously been found to be at increased risk for developing severe dengue and to have higher mortality rates. Therefore there is limited novelty in these findings and it is unclear what additional benefit they would add to support clinical decision making. 

Additional experiments: 

• Given the strong correlation with age with disease severity, the authors should consider subsampling their cohort by age and evaluating whether there are specific associations with severe dengue in adults only and children only. This may aid in generating more precision approaches to predicting severe dengue. 

• The authors used the 2009 WHO criteria to classify patients. These criteria include uncomplicated dengue, dengue with warning signs (an intermediate classification which has higher likelihood of progressing to severe dengue) and severe dengue. If the authors have the data evaluating the clinical and laboratory findings in uncomplicated dengue patients and dengue with warning signs patients at admission who then progressed to severe dengue during their hospitalization would add to predictive power of their findings. 

• Since the authors used the 2009 criteria to diagnose severe dengue which include organ impairment as part of the criteria it’s unclear how these patients may have affected outcomes as patients with plasma leakage/hemorrhage and organ damage may have two distinct immuno-pathogeneses. Separating the severe dengue patients by syndrome may further clarify the association of plasma leakage and hemorrhage with severity.

PLOS authors have the option to publish the peer review history of their article (what does this mean?). If published, this will include your full peer review and any attached files.

Reviewer #1: No

Reviewer #2: Yes: Camila D. Odio

Reviewer #3: No
---

## [Decision Letter · Decision Letter 1]

10 Feb 2023

Dear Dr. Hossain,

We are pleased to inform you that your manuscript 'Demographic characteristics, clinical symptoms, biochemical markers and probability of occurrence of severe dengue:A multicenter hospital-based study in Bangladesh' has been provisionally accepted for publication in PLOS Neglected Tropical Diseases.

Best regards,

Marc Choisy

Academic Editor

Abdallah Samy

Section Editor

The three reviewers and myself think that you did a very deep revision of your manuscript that successfully addressed all the weakness of the first version, both on the content and the form.

Reviewer's Responses to Questions

**Key Review Criteria Required for Acceptance?**

**Methods**

-Are the objectives of the study clearly articulated with a clear testable hypothesis stated?

-Is the study design appropriate to address the stated objectives?

-Is the population clearly described and appropriate for the hypothesis being tested?

-Is the sample size sufficient to ensure adequate power to address the hypothesis being tested?

-Were correct statistical analysis used to support conclusions?

-Are there concerns about ethical or regulatory requirements being met?

Reviewer #1: All queries were addressed in the revised draft

Reviewer #2: (No Response)

**Results**

-Does the analysis presented match the analysis plan?

-Are the results clearly and completely presented?

-Are the figures (Tables, Images) of sufficient quality for clarity?

Reviewer #1: Results are presented according to the analysis plan

Reviewer #2: (No Response)

**Conclusions**

-Are the conclusions supported by the data presented?

-Are the limitations of analysis clearly described?

-Do the authors discuss how these data can be helpful to advance our understanding of the topic under study?

-Is public health relevance addressed?

Reviewer #1: Conclusions were drawn based on the data

Reviewer #2: (No Response)

**Editorial and Data Presentation Modifications?**

Reviewer #1: (No Response)

Reviewer #2: (No Response)

**Summary and General Comments**

Reviewer #1: (No Response)

Reviewer #2: The authors' revisions were thorough and helpful. I now understand that they are using the models to differentiate among the markers of severe dengue and identify which ones are most strongly associated with severe dengue. I think this paper is a useful addition to the literature on the clinical presentations of severe dengue and may be beneficial to clinical providers.

PLOS authors have the option to publish the peer review history of their article (what does this mean?). If published, this will include your full peer review and any attached files.

Reviewer #1: No

Reviewer #2: **Yes: **Camila D. Odio

---

## [Editor Report · Acceptance letter]

9 Mar 2023

Dear Dr. Hossain,

We are delighted to inform you that your manuscript, "Demographic characteristics, clinical symptoms, biochemical markers and probability of occurrence of severe dengue:A multicenter hospital-based study in Bangladesh," has been formally accepted for publication in PLOS Neglected Tropical Diseases.

Best regards,

Shaden Kamhawi

co-Editor-in-Chief

Paul Brindley

co-Editor-in-Chief
